# Differential Response of Macrobenthic Abundance and Community Composition to Mangrove Vegetation

**Sin-He Pan** [1,†], **Chuan-Wen Ho** [1,†], **Chiao-Wen Lin** [1], **Shou-Chung Huang** [2] **and Hsing-Juh Lin** [1,*]

1. Department of Life Sciences and Innovation and Development Center of Sustainable Agriculture, National Chung Hsing University, Taichung 40227, Taiwan; a5487148916748ohyeah@gmail.com (S.-H.P.); cw.ho@nchu.edu.tw (C.-W.H.); clin10@ncsu.edu (C.-W.L.)
2. Taiwan Wetland Society, No. 33, Guohua Street, Zhubei City 30244, Taiwan; schuang.120@gmail.com
* Correspondence: hjlin@dragon.nchu.edu.tw; Tel.: +886-4-2284-0416 (ext. 510)
† S.-H.P. and C.-W.H. contributed equally to the work.

**Abstract:** The mass planting of mangroves has been proposed as a mitigation strategy to compensate for mangrove loss. However, the effects of mangrove vegetation on the abundance and community composition of macrobenthos remain controversial. The macrobenthic communities in four intact mangrove forests with different conditions and the adjacent nonvegetated mudflats of two mangrove species with distinct stand structures on the western coast of Taiwan were examined. Some macrobenthic taxa occurred only in the mangroves, suggesting macrobenthic critical habitats. Seasonal shift in community composition was more pronounced in the mudflats than in the mangroves, possibly due to the rich food supply, low temperature, and shelter function provided by mangrove forests. However, crab density was always lower in the mangroves than in the mudflats. There was a negative relationship between the stem density of *Kandelia obovata* (S., L.) and infaunal density. The pneumatophore density of *Avicennia marina* (Forsk.) correlated negatively with epifaunal density. Our results show that the response of macrobenthic abundance and community composition to mangrove vegetation was inconsistent. We reason that mangroves are critical habitats for the macrobenthos in the mudflats. However, if mangrove tree density is high, we predict that the macrobenthic density will decrease. This suggests that at some intermediate level of mangrove tree density, where there are enough mangrove trees to harbor a macrobenthic community but not enough trees to significantly reduce this density, mangroves management can be optimally achieved to promote the presence of a diverse and dense macrobenthic community.

**Keywords:** *Avicennia marina*; *Kandelia obovata*; macrobenthos; pneumatophore; tree density



## 1. Introduction

Mangroves are marine nurseries for juvenile fish [1,2] and habitats for a variety of benthic [3] and economically important species [4]. Unfortunately, mangroves are among the most threatened tropical ecosystems in the world due to human disturbance, coastal construction [5,6], and climate change [7–10]. The loss of mangrove area globally was estimated to be 0.22% year$^{-1}$ [11].

Macrobenthos are benthic animals retained by a 0.5-mm sieve, either living on the surface of a substrate (epifauna) or buried or burrowing in the sediment (infauna), according to their habitat [12]. Mangrove macrobenthos not only may serve as an important link between recalcitrant detritus and consumers at higher trophic levels [1] but also indirectly affect the biogeochemical cycle through changing soil permeability, oxidation, and water content [13–15]. Furthermore, they are also a critical connection along the aquatic continuum between streams and oceans [16] and may bioaccumulate or biomagnify metals through food chain [17]. However, the effects of mangrove vegetation on the macrobenthic community remain controversial. The roots and pneumatophores of trees can improve the aeration of mangrove sediments [18] and intercept detritus and litterfall, which can

increase the organic content in sediments [19]. A higher organic content in the sediments generally leads to a higher abundance and richness of macrobenthos in mangroves [20,21]. The stand structure of mangroves may enhance habitat complexity, provide hard substrates for epifauna [22], and reduce the risk of washout by outgoing tidal currents [23]. Epiphytes growing on the surface of a stand structure may become food sources for epifauna [24]. The belowground structure can provide shelter for infauna, allowing them to hide from predators [23]. The oxic layer surrounding mangrove roots can increase the abundance of microbes, which may become a food source for polychaetes [25]. A higher density of crabs is associated with a higher mangrove stem density [26]. The species diversity of the macrobenthic community is higher in mangroves than in adjacent mudflats [27].

However, mangrove stand structure may compact the sediment and cause a reduction in water and oxygen contents [28]. The increased bulk density of sediment may become a barrier to the burrowing, movement, and feeding of infauna [19,29], which may also increase their predation risk [30]. Mangrove vegetation may hinder the displays and lekking behavior of fiddler crabs [31]. Dense mangrove canopies can reduce the irradiance that reaches the sediments under mangroves, which may decrease the production of benthic algae and feeding by grazers [32,33]. It is expected that mangrove vegetation can result in a distinct microhabitat from nonvegetated intertidal flats, which may result in a shift in the abundance and community composition of macrobenthos. This is more relevant than ever, as the mass planting of mangroves has been proposed as a mitigation strategy to compensate for mangrove loss globally [34].

Different mangrove species may possess distinct stand structures, which may impose different effects on macrobenthos. For example, *Kandelia obovata* (S., L.) possesses prop roots, whereas *Avicennia marina* (Forsk.) is surrounded by pneumatophores. The main objectives in this study are to (1) characterize the sediment features and the abundance and community composition of macrobenthos in the mangroves of two species (*K. obovata* and *A. marina*) and adjacent nonvegetated mudflats and (2) compare the sediment features and the abundance and community composition of macrobenthos between mangroves and nonvegetated mudflats. We hypothesized that the sediment organic content and the density and species richness of macrobenthos are higher in the mangroves of both species than in the adjacent nonvegetated mudflats.

## 2. Materials and Methods

### 2.1. Study Sites

Mangroves are widely distributed on the western coast of Taiwan. While *Kandelia obovata* is dominant on the northern (subtropical) coast, *Avicennia marina* is dominant on the southern (tropical) coast [35]. There were four study sites in this study: two (Xinfeng, XF and Zhunan, ZN) were located in *K. obovata* mangroves, and two (Budai, BD and Beimen, BM) were located in *A. marina* mangroves (Supplementary Figure S1). These mangroves generally experience a semidiurnal tide with a tidal range of approximately 2 m [13]. From 2018 to 2019, the mean air temperature was 15 °C in winter and 30 °C in summer; the annual precipitation was approximately 1700 mm (Central Weather Bureau, Taiwan)

The age of these riverine forests ranged from 40 to 120 years (Table 1), and all of them were introduced or planted. Mangroves cover ranged from 8.25 ha in XF to 19.5 ha in BD. The forest still expands in ZN and BD. While the forest in XF is shrinking due to sediment erosion, the forest in BM is threatened by fishery and agriculture activity. The distance to the sea was relatively far in ZN (1.68 km) and short in BD (0.40 km). The submersion time during high tide was shortest in ZN, indicating that the elevation in ZN was higher than that in the other sites. The sediment salinity was lower (0.93 psu) at XF than at the other sites, possibly due to the frequent flooding of stream water.

**Table 1.** Forest structure and sediment features (mean $\pm$ SD) of the four mangrove sites on the western coast of Taiwan. The different superscripted letters indicate significant differences ($p < 0.05$), as determined by one-way ANOVA or the Kruskal–Wallis test. NA: data not available.

| Site | XF | ZN | BD | BM |
|---|---|---|---|---|
| Latitude | 24°54′ N | 24°40′ N | 23°21′ N | 23°17′ N |
| Longitude | 120°58′ E | 120°50′ E | 120°07′ E | 120°06′ E |
| Dominant tree species | *Kandelia obovata* (S., L.) | *Kandelia obovata* | *Avicennia marina* (Forsk.) | *Avicennia marina* |
| Forest age (year) | 70 | 40 | 120 | 100 |
| Forest condition | Intact, eroded | Intact, expanded | Intact, expanded | Intact, threatened |
| Cover area (ha) | 8.25 | 14.5 | 19.5 | 18.9 |
| Distance to the sea (km) | 0.70 | 1.68 | 0.40 | 0.62 |
| Submersion time during a high tide in June 2020 (minutes) | 145 | 35 | 502 | 452 |
| $NO_2^- + NO_3^-$ ($\mu$M) [#] | 320 $\pm$ 66 [c] | 28.3 $\pm$ 5.8 [b] | 6.36 $\pm$ 1.50 [a] | 15.3 $\pm$ 3.6 [b] |
| $NH_4^+$ ($\mu$M) [#] | 66.3 $\pm$ 14.1 [b] | 100 $\pm$ 22 [b] | 6.80 $\pm$ 1.38 [a] | 16.1 $\pm$ 3.9 [a] |
| $PO_4^{-3}$ ($\mu$M) [#] | 30.4 $\pm$ 7.4 [b] | 24.5 $\pm$ 9.8 [b] | 2.83 $\pm$ 0.56 [a] | 2.92 $\pm$ 0.65 [a] |
| Tree height (m) | 5.20 $\pm$ 2.09 [c] | 5.04 $\pm$ 1.58 [c] | 4.02 $\pm$ 2.86 [b] | 3.21 $\pm$ 1.95 [a] |
| Diameter at breast height (DBH, cm) | 6.00 $\pm$ 0.10 [b] | 6.13 $\pm$ 0.07 [b] | 5.36 $\pm$ 0.05 [a] | 7.16 $\pm$ 0.08 [b] |
| Tree density (ind. m$^{-2}$) | 2.30 $\pm$ 0.27 [b] | 1.80 $\pm$ 0.11 [b] | 0.60 $\pm$ 0.16 [a] | 0.33 $\pm$ 0.07 [a] |
| Pneumatophore density (ind. m$^{-2}$) | NA | NA | 119.11 $\pm$ 13.83 [a] | 244.81 $\pm$ 30.44 [b] |
| Sediment light intensity ($\mu$mol photon m$^{-2}$ s$^{-1}$) | 65.2 $\pm$ 82.6 [a] | 124.2 $\pm$ 109.8 [b] | 185.6 $\pm$ 260.3 [b] | 189.6 $\pm$ 104.6 [b] |
| Sediment salinity | 0.93 $\pm$ 0.17 [a] | 2.64 $\pm$ 0.26 [b] | 3.43 $\pm$ 0.47 [c] | 4.25 $\pm$ 0.61 [c] |
| Sediment water content (%) | 28.99 $\pm$ 5.21 [a] | 26.08 $\pm$ 5.32 [a] | 50.38 $\pm$ 8.03 [c] | 34.60 $\pm$ 3.22 [b] |
| Sediment organic content (%) | 4.78 $\pm$ 0.76 [c] | 4.08 $\pm$ 0.13 [b] | 7.60 $\pm$ 0.66 [d] | 3.49 $\pm$ 0.37 [a] |
| Sediment median grain size (mm) | 0.061 $\pm$ 0.013 [c] | 0.024 $\pm$ 0.003 [a] | 0.032 $\pm$ 0.003 [b] | 0.037 $\pm$ 0.009 [b] |
| Sediment sorting coefficient | 1.90 $\pm$ 0.10 [c] | 1.40 $\pm$ 0.05 [a] | 2.25 $\pm$ 0.06 [d] | 1.58 $\pm$ 0.12 [b] |
| Sediment bulk density (g cm$^{-3}$) | 1.03 $\pm$ 0.08 [b] | 1.28 $\pm$ 0.09 [c] | 0.57 $\pm$ 0.15 [a] | 0.96 $\pm$ 0.13 [b] |
| Sediment silt/clay content (%) | 41.69 $\pm$ 6.74 [a] | 88.72 $\pm$ 2.66 [d] | 61.73 $\pm$ 1.26 [b] | 66.68 $\pm$ 5.78 [c] |
| Sediment ORP (mV) | 174.9 $\pm$ 73.9 [b] | 69.3 $\pm$ 83.6 [b] | −258.5 $\pm$ 68.4 [a] | −162.1 $\pm$ 45.1 [a] |
| Benthic chlorophyll *a* concentration (mg m$^{-2}$) | 65.60 $\pm$ 5.40 [b] | 30.28 $\pm$ 14.61 [a] | 131.19 $\pm$ 54.48 [c] | 68.08 $\pm$ 21.25 [b] |

[#] Water nutrient data were derived from Wu (2020) [36].

## 2.2. Macrobenthic Sampling

Macrobenthos in vegetated mangroves (V) were sampled in February (winter), April (spring), July (summer), and October (autumn) from 2018 to 2019 for two complete seasonal cycles. There were three random plots (5 m × 5 m for each plot) in each mangrove and three replicate samples in each plot for each season. Sampling in the adjacent nonvegetated (NV) mudflats was also conducted in three random plots for each season during the daytime at low tide in 2019 only. These plots within each site were separated by at least 3 m. To reduce edge effects, the distance of the plots to the forest edge was >100 m.

Infauna in the sediments were collected by pushing a sampling core (LY082.1500, Zhen-Yong Industrial Co., Ltd., Taichung, Taiwan) with a diameter of 10 cm into the sediment to a depth of 10 cm in each plot. All infaunal samples were sieved through a 0.5-mm screen. All epifauna on the surface of the sediment in each plot were also collected. Both the epifauna and infauna samples were fixed with 95% alcohol and a few drops of menthol and brought back to the laboratory for identification and counting.

The infauna and epifauna were identified to the lowest possible taxonomic level according to Chinese Polychaetes [37], Guide to Polychaetes (Annelida) in Qatar Marine Sediments [38], the Crustacean Fauna of Taiwan [39], Mangroves of Taiwan [40], and the Guide to Taiwanese Shells [41]. To avoid overestimating the abundance of macrobenthos, only those mollusks with soft body parts were quantified. Arthropoda, Oligochaeta, Polychaeta, and Sipuncula were counted only when the cephalic portion was preserved. All the abundance data were standardized and transformed into density data by dividing the count number by the sampling surface area and are expressed in terms of individual number per m$^2$.

The density of crabs was also estimated before each macrobenthic sampling by the visual counting method of Skov and Hartnoll [42]. At least two individual researchers stood 3.5 m away from each plot and counted the number of crabs in the plots (1 × 1 m,

n = 3 squares per site for each season and a 10-min observation period for each square) within the mangroves or mudflats with binoculars (50 × 12 mm, Nikon, Japan). The density of crabs was counted at spring low tide during the daytime to ensure maximum crab activities.

### 2.3. Sediment Features

After collecting samples of the macrobenthos, the photosynthetically active radiation (PAR, 400–700 nm) under the mangrove canopy (n = 6) and on the mudflats (n = 3) was determined by a quantum meter (Li-1400, LI-COR, Lincoln, NE, USA). Duplicate sediment samples were randomly collected from the top 5 cm in each plot in a plastic tube with a diameter of 2.9 cm and pooled to form a single sample for each plot for further analyses in the laboratory. A portion of the sediments was dried at 60 °C to a constant weight, and the percentage of water loss was calculated as the water content and the bulk density [43]. The dried sediments were further combusted at 500 °C for 4 h, and the percentage of weight loss was calculated as the organic matter content. For granulometric analysis, the median grain size, silt/clay content, and sorting coefficient of the sediments were determined following the modified pipette method [44]. A sediment sample from the top 10 cm was also taken from each plot with a sampling corer (LY082.1500, Zhen-Yong Industrial Co., Ltd., Taichung, Taiwan) with a diameter of 10 cm, and the sediment temperature and oxidation-reduction potential (ORP) in the center of the core (i.e., at a depth of 5 cm) were immediately measured with an ORP meter (ORP30, CLEAN L'eau Instruments Co., Ltd., Taoyuan, Taiwan).

For collecting benthic microalgae, plastic cores (inner diameter of 1 cm, n = 9, in each plot in the mangroves and mudflats) were pushed into the sediments to a depth of 1 cm, where benthic microalgae are mainly located. The benthic chlorophyll *a* (Chl *a*) concentration was determined by the acetone method [45] in the laboratory and was used as an index of the biomass of the benthic microalgae.

### 2.4. Mangrove Forest Structure

After collecting the macrobenthic samples and determining the sediment features, the number of stems was counted, and the diameter at breast height (DBH, cm) of all the stems of *A. marina* and *K. obovata* was measured in the three plots at each site. The number of pneumatophores of *A. marina* was also counted in triplicate squares (each square: 1 m × 1 m) randomly placed in each plot.

### 2.5. Statistical Analyses

Prior to statistical analysis, all the data were analyzed by a Shapiro–Wilk test to assess whether they were normally distributed and by Levene's test to assess whether the variances were homogeneous. If the data were not normally distributed, they were transformed according to the suggestions of Clarke and Warwick [46]. Transformations were necessary for the following data: (a) tree density, DBH, light intensity, and crab density (square root); (b) organic matter content, median grain size, sediment temperature, benthic Chl *a* concentration, and the Shannon–Wiener diversity of the macrobenthic community (fourth root); and (c) pneumatophore density, sediment water content, and macrobenthic density (log). Either Student's *t*-test or the Wilcoxon rank-sum test (if the data still showed heterogeneity of variance after transformation) was applied to examine the difference in macrobenthic variables between the mangroves and mudflats at each site. Either two-way ANOVA or the Kruskal–Wallis test (if the data still showed heterogeneity of variance after transformation) was applied to examine whether there were seasonal and site variations in macrobenthic variables in the mangroves. Tukey's HSD (honestly significant difference) test was further applied for multiple comparisons if ANOVA showed significant differences. Dunn's multiple comparison test was applied for multiple comparisons if the Kruskal–Wallis test showed a significant difference. The analyses described above were conducted in SigmaPlot version 12.5.

The most influential factors in the sediments of the four mangroves and adjacent mudflats were determined by principal component analysis (PCA). The macrobenthic community data were log-transformed to downweight the influence of the dominant taxa before analysis [47]. Bray–Curtis similarity analysis was used to generate a resemblance matrix of the log-transformed community data, and pairwise tests were used to determine the differences between sites and among habitats and seasons. Resemblance matrices were used to display the similarity of the communities in the different mangroves in different seasons by nonmetric multidimensional scaling (MDS). Similarity of percentage (SIMPER) was applied to determine the most common taxa in the macrobenthic samples at each site. PERMDISP was used to test the difference in dispersion between the macrobenthic community sampled in the mangroves and adjacent mudflats at each site. The relationship between the macrobenthic community and sediment features was further analyzed by a distance-based linear model (DistLM). A distance-based redundancy analysis (dbRDA) ordination diagram was then used to visualize the fitted models. The abovementioned statistical analyses were conducted using PRIMER 6.1.13 and PERMANOVA+ [48,49].

## 3. Results

### 3.1. Mangrove Forest Structure

The average height of *K. obovata* was approximately 5 m, while that of *A. marina* was 3–4 m (Table 1). The DBH of the mangrove trees ranged from 5–7 cm. Tree density was significantly higher in the *K. obovata* mangroves than in the *A. marina* mangroves. The pneumatophore density of *A. marina* was two times higher in BM than in BD. Light intensity was significantly lower in the mangroves than in mudflats across all the sites (Kruskal–Wallis test, $p < 0.001$). Light intensity and sediment temperature were significantly lower in mangroves in XF than in the other mangroves, possibly due to the relatively high tree density.

### 3.2. Sites Response

A total of 66 taxa, including Actiniaria, Amphipoda, Bivalvia, Decapoda, Diptera, Gastropoda, Isopoda, Nemertea, Oligochaeta, Perciformes, Polychaeta, Sessilia, Sipuncula, Tanaidacea, and Xiphosurida, were identified (Supplementary Table S1). In BM, the taxon richness sampled in the mangrove forests was significantly higher than that in the mudflats (Supplementary Table S2); however, in ZN, the taxon richness was significantly lower in the mangrove forests than in the mudflats. The Shannon–Weiner diversity (H′) of the macrobenthic communities collected in the mangroves averaged from 0.48 to 2.30. The spatial pattern of H′ was consistent with the pattern of macrobenthos taxon richness in ZN and BM. There was no significant difference in the taxon richness and H′ between the mangrove forests and mudflats in XF and BD. The taxon richness and H′ in the mangrove forests in BM were significantly higher than the richness and H′ in the other mangrove forests (Supplementary Table S3). Nevertheless, no significant differences in the richness and H′ were detected among the mudflats (Supplementary Table S4).

The density of macrobenthos in the mangrove forests averaged from 226 to 6830 ind. m$^{-2}$, whereas the density of macrobenthos in the mudflats averaged from 466 to 9821 ind. m$^{-2}$. The density of macrobenthos in the mangrove forests was significantly different from that in the mudflats in ZN and BM but not in XF or BD (Supplementary Table S2). In ZN, the density of macrobenthos was significantly lower in the mangrove forests than in the mudflats (Figure 1); however, in BM, the density was higher in the mangrove forests than in the mudflats (Figure 2).

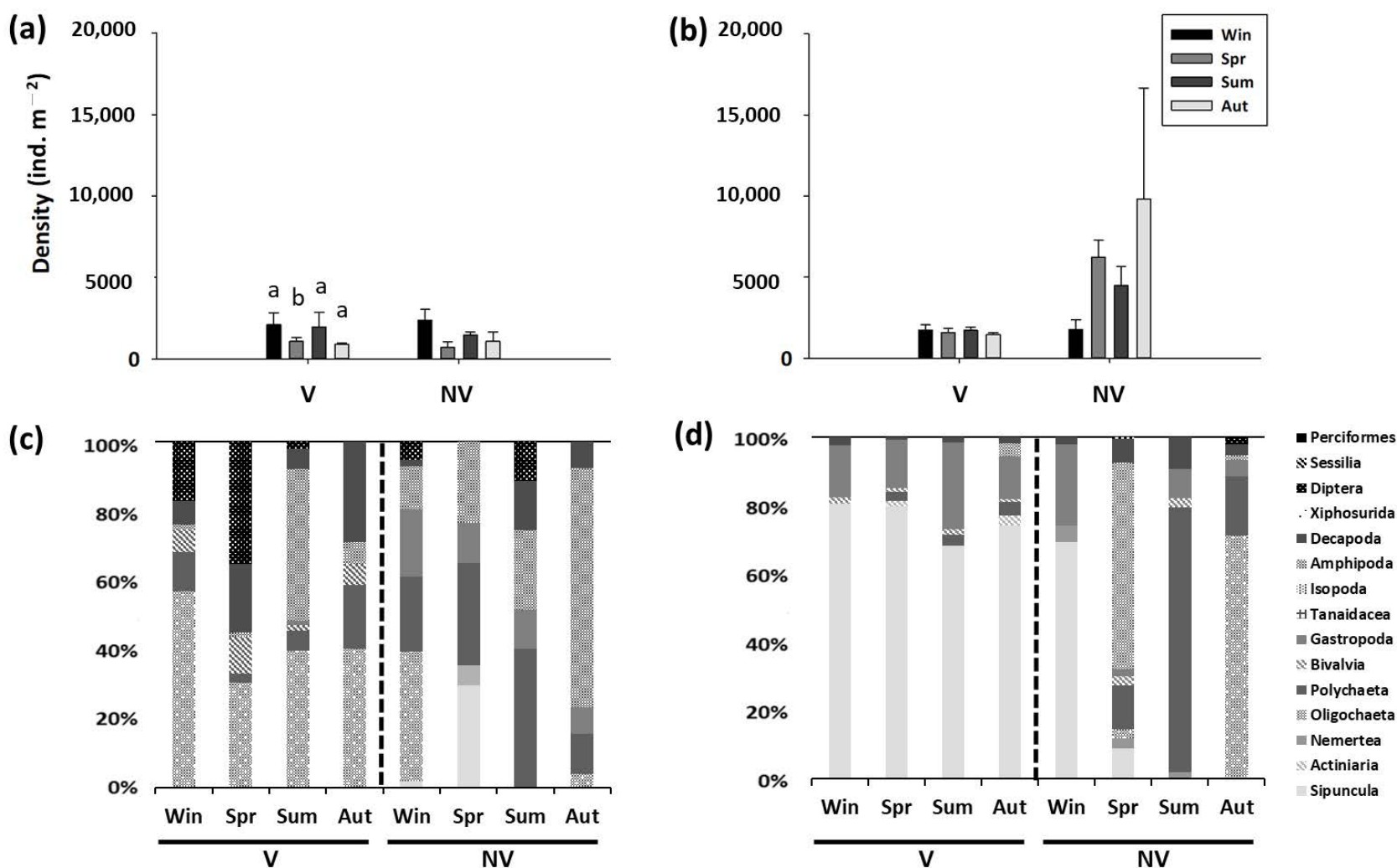

**Figure 1.** Seasonal variations in the density (mean ± SE) and macrobenthic community composition in the *Kandelia obovata* (S., L.) mangroves in (**a,c**) XF and (**b,d**) ZN. v, mangroves; nv, nonvegetated mudflats; win, winter; spr, spring; sum, summer; aut, autumn. Within each site, different letters above columns denote significant (*p* < 0.05) differences among seasons with the Tukey's post-hoc test.

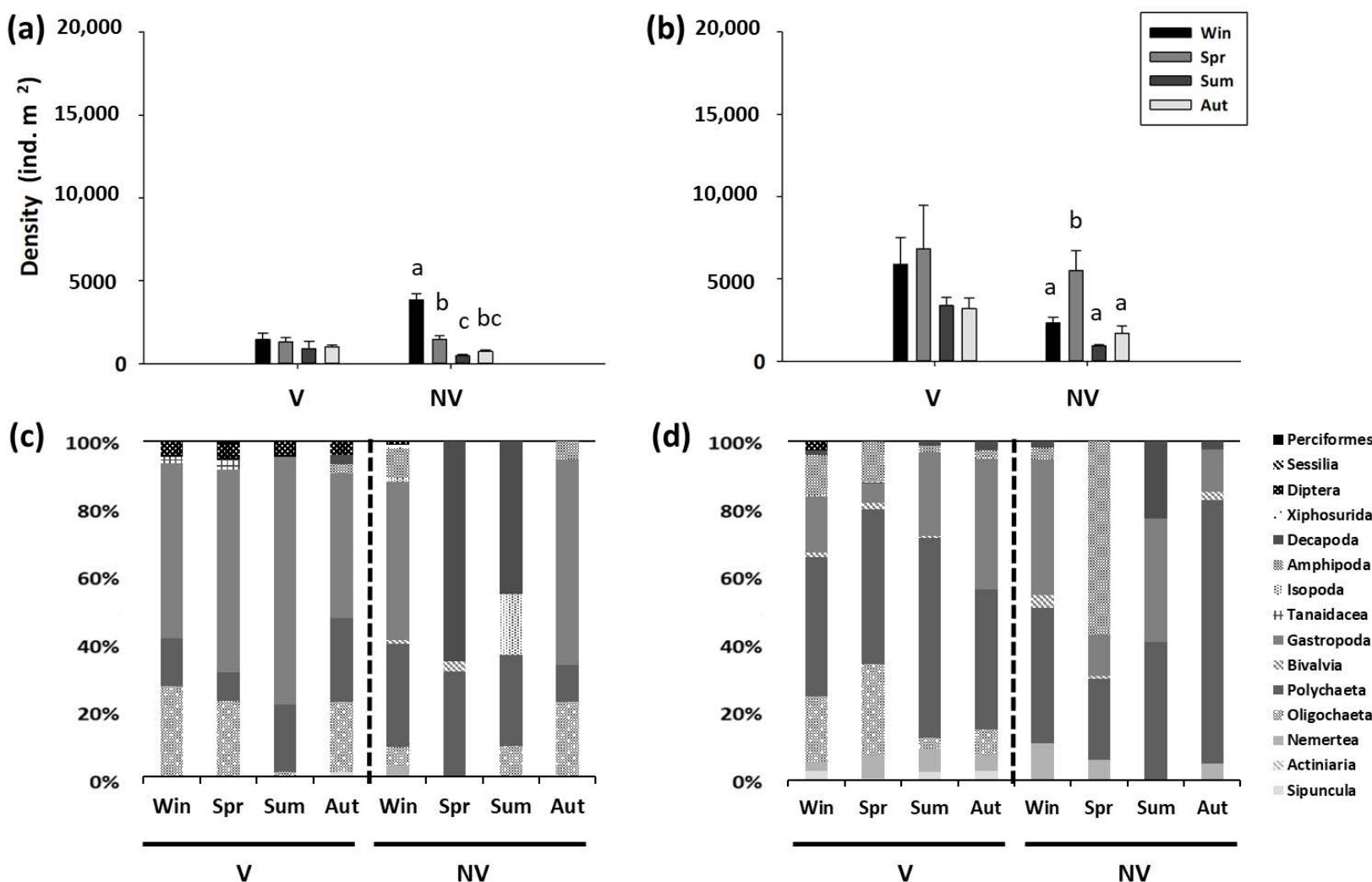

**Figure 2.** Seasonal variations in the density (mean ± SE) and macrobenthic community composition in the *Avicennia marina* mangroves in (**a**,**c**) BD and (**b**,**d**) BM. v, mangroves; nv, nonvegetated mudflats; win, winter; spr, spring; sum, summer; aut, autumn. Within each site, different letters above columns denote significant ($p < 0.05$) differences among seasons with the Tukey's post-hoc test.

There was a significant site effect on the density of macrobenthos in the mangrove forests and mudflats (Supplementary Tables S3 and S4). The density was lowest in the frequently flooded site (XF), where the community was dominated by Oligochaeta and Amphipoda (Figure 1); the density was highest in the mangrove forests and mudflats of BM, where the dominant taxon was Polychaeta (Figure 2). While the macrobenthic community at the higher elevation site (ZN) was dominated by Sipuncula and Decapoda, Gastropoda and Polychaeta were the dominant taxa at the lower elevation site (BD).

The crab density in the mangroves averaged from 0.0 to 4.3 ind. $m^{-2}$, whereas the crab density in the mudflats averaged from 0.0 to 56.3 ind. $m^{-2}$ (Figure 3). The crab density was significantly lower in all the mangrove forests than in the mudflats (Supplementary Table S2).

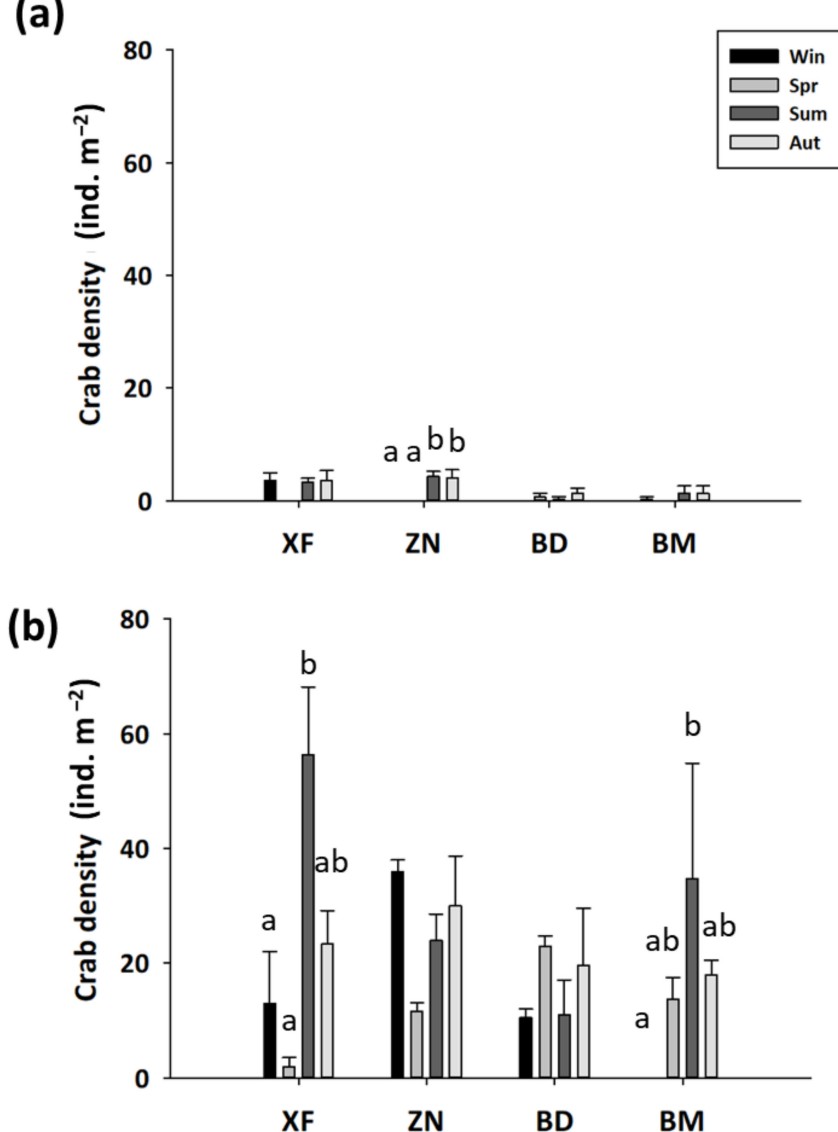

**Figure 3.** Crab density (mean ± SE) in the (**a**) mangroves and (**b**) nonvegetated mudflats of the four sites on the western coast of Taiwan in different seasons. win, winter; spr, spring; sum, summer; aut, autumn. Within each site, different letters above columns denote significant ($p < 0.05$) differences among seasons with the Tukey's post-hoc test.

### 3.3. Seasonal Response

The seasonal patterns of the macrobenthic community composition in the mangrove forests were more consistent than those in the mudflats across all the sites. While the polychaetes Lumbrineridae; the bivalves *Potamocorbula laevis*, Cyrenidae, and *Glauconome chinensis*; the gastropods *Iravadia quadrasi* and *Hyala bella*; tanaids (Tanaidacea); and the amphipod Aoridae existed only in the mangrove forests, the gastropods *Batillaria zonalis* and *Pirenella cingulata* were present only in the mudflats (Table 2).

**Table 2.** SIMPER analysis of the macrobenthic communities collected from mangrove forests and nonvegetated mudflats on the western coast of Taiwan. The top 5 common and dominant taxa in each habitat were ranked by a contribution > 5% to the percentage similarity within each community.

| XF | ZN | BD | BM |
|---|---|---|---|
| Mangrove forests | | | |
| Oligochaeta 29.16% | *Phascolosoma arcuatum* 85.28% | Stenothyridae 43.27% | Capitellidae 21.62% |
| Decapoda juveniles 25.36% | Assimineidae 8.98% | *Iravadia quadrasi* 26.84% | Ampharetidae 16.89% |
| Dolichopodidae larvae 20.39% | | Ampharetidae 7.91% | Nemertina 16.64% |
| Corophiidae 7.84% | | Oligochaeta 6.44% | *Iravadia quadrasi* 12.67% |
| Capitellidae 6.76% | | Dolichopodidae larvae 5.56% | Oligochaeta 10.76% |
| Nonvegetated mudflats | | | |
| Corophiidae 25.10% | Capitellidae 21.21% | *Mictyris brevidactylus* 20.77% | Capitellidae 37.66% |
| Capitellidae 24.52% | Assimineidae 20.57% | *Pirenella cingulata* 19.91% | Naticidae 14.75% |
| Assimineidae 18.78% | *Metaplax longipes* 13.43% | Oligochaeta 15.79% | Nemertina 11.59% |
| Dolichopodidae larvae 12.76% | *Phascolosoma arcuatum* 9.52% | *Pirenella alata* 14.79% | Cossuridae 10.91% |
| Oligochaeta 5.87% | Corophiidae 9.13% | Decapoda juveniles 10.50% | Spionidae 8.25% |

The crab density (Table 3) in the mangrove forests was significantly lower in spring but was higher in summer and autumn (Supplementary Table S3). There was no significant difference in crab density among the mangrove forests. In the mudflats, there was a significant interaction effect of site and season on crab density (Supplementary Table S4).

**Table 3.** Density of crab species (ind. m$^{-2}$) in the mangrove forests (v) and nonvegetated mudflats (nv) at the four sites on the western coast of Taiwan. The bold values indicate the most 2–3 dominant species in each habitat.

| Species | XF | | ZN | | BD | | BM | |
|---|---|---|---|---|---|---|---|---|
| | v | nv | v | nv | v | nv | v | nv |
| *Mictyris brevidactylus* | 0.00 | 0.00 | 0.00 | 0.00 | 0.00 | 4.36 | 0.00 | 0.00 |
| Sesarmidae | 1.75 | 1.36 | 1.75 | 1.00 | 0.58 | 0.00 | 0.50 | 0.09 |
| *Helice formosensis* | 0.00 | 0.64 | 0.00 | 0.00 | 0.00 | 0.00 | 0.00 | 0.00 |
| *Macrophthalmus* sp. | 0.00 | 1.09 | 0.00 | 14.36 | 0.00 | 0.00 | 0.17 | 18.00 |
| *Austruca lactea* | 0.33 | 17.82 | 0.08 | 0.55 | 0.00 | 8.64 | 0.00 | 0.00 |
| *Austruca triangularis* | 0.00 | 0.00 | 0.00 | 0.00 | 0.00 | 0.00 | 0.08 | 0.00 |
| *Austruca perplexa* | 0.00 | 0.09 | 0.00 | 0.00 | 0.00 | 0.00 | 0.00 | 0.00 |
| *Gelasimus borealis* | 0.00 | 0.00 | 0.00 | 0.00 | 0.00 | 1.09 | 0.00 | 0.00 |
| *Tubuca arcuata* | 0.50 | 3.45 | 0.08 | 1.00 | 0.00 | 1.82 | 0.00 | 0.00 |
| *Xeruca formosensis* | 0.08 | 0.18 | 0.17 | 0.00 | 0.00 | 0.00 | 0.00 | 0.00 |
| *Ilyoplax* sp. | 0.00 | 0.00 | 0.00 | 0.45 | 0.00 | 0.00 | 0.00 | 0.00 |
| *Scopimera* sp. | 0.00 | 0.00 | 0.00 | 0.00 | 0.00 | 0.18 | 0.00 | 0.00 |
| *Tmethypocoelis ceratophora* | 0.00 | 0.00 | 0.00 | 7.09 | 0.00 | 0.45 | 0.00 | 0.00 |

### 3.4. Environmental Variables Response

The sediment water content was associated with substrate elevation (Table 1). It was higher in BD due to its lower elevation and longer submersion time and lower in ZN due to the higher elevation and shorter submersion time. Correspondingly, the sediment bulk density was higher in ZN but was lower in BD. The sediment sorting coefficient was higher (very poorly sorted) in BD and lower (poorly sorted) in ZN. The benthic Chl *a* concentration was also associated with the water content and was higher in BD and lower in ZN. The median grain size in the mangrove sediments averaged 0.024~0.061 mm, indicating that the sediments varied from silt to very fine sand. The grain size was smaller at the high-elevation site (ZN) and larger at the frequently flooded site (XF). The silt/clay content was also higher in ZN and lower in XF. The sediment organic content was higher in BD and lower in BM.

PCA was used to distinguish the sediment features of the four mangrove forests and mudflats. The results of PCA showed that the first and second axes together explained 63.5% of the variance in the data (Figure 4). The first axis accounted for 41.1% of the variance; bulk density, water content, organic content, grain size, and slit/clay content were the most correlated variables along this axis. The second axis accounted for the remaining 22.4% of the variance; light intensity, sediment temperature, benthic Chl *a* concentration, and ORP were the most correlated variables along this axis. The sediment features in the mangrove forests and mudflats can be separated on the combined first and second axes. While all the mangrove forests had higher organic matter and silt/clay contents, all the mudflats had higher light intensity and sediment temperature and larger grain size. In addition, the sediment ORP and bulk density were higher in the *K. obovata* mangroves, but the water content and benthic Chl *a* concentration were higher in the *A. marina* mangroves. There was no clear seasonal pattern that could be detected in the sediment features across the study sites.

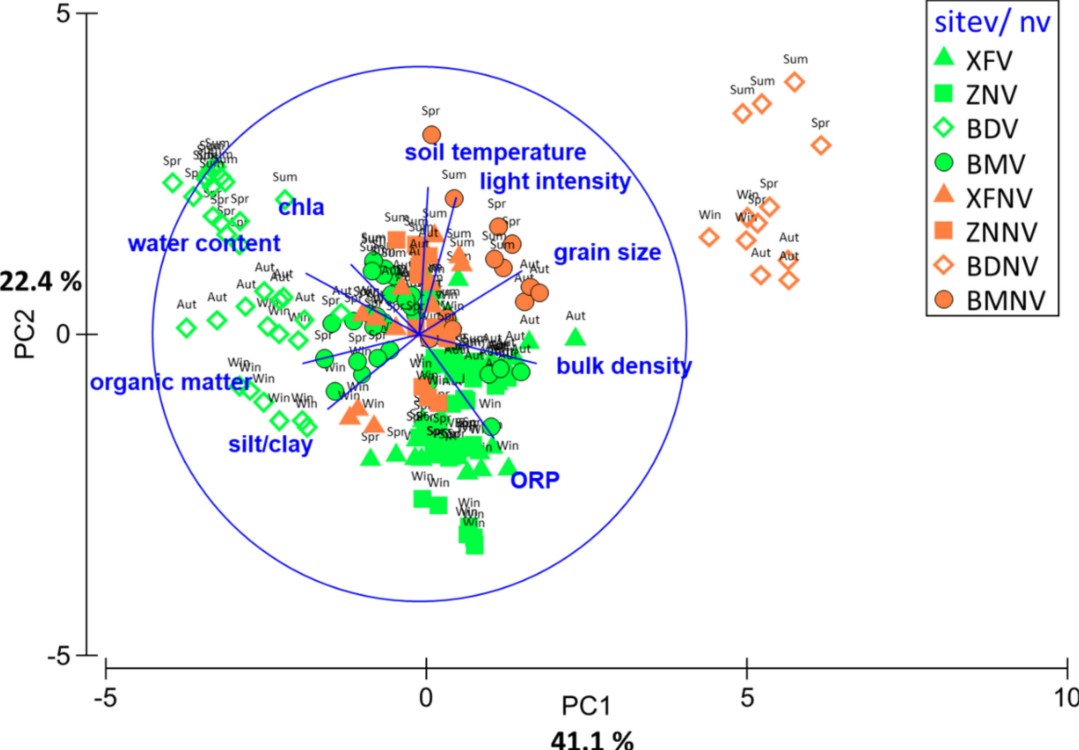

**Figure 4.** PCA of the sediment features in mangroves (V) and nonvegetated mudflats (NV) on the western coast of Taiwan in different seasons. Black: mangroves, gray: mudflats, ▲: XF, ■: ZN, ◇: BD, ●: BM. win, winter; spr, spring; sum, summer; aut, autumn.

To better understand the factors regulating the macrobenthic communities in the *K. obovata* and *A. marina* mangroves, the data collected in this study were combined with those of the macrobenthic communities in other mangroves in Taiwan and analyzed with a DistLM and by dbRDA. The data included were collected from the macrobenthic communities in the *K. obovata* mangroves in Zhuwei (ZW) [50] and Wazihwei (WZ) [51] in northern Taiwan and Fangyuan (FY) [52] in central Taiwan and from the *A. marina* mangroves in Fangyuan (FY) [52] and Chiku (CK) [53] in southern Taiwan (Supplementary Figure S1).

The results of the DistLM and dbRDA showed that the first two axes can explain 33.5% of the total variation in the macrobenthic communities in the *K. obovata* mangroves (Figure 5a). The macrobenthic communities collected in ZW were similar to those collected in WZ and FY. The first axis, which correlated most strongly with the silt/clay content (r = −0.898) and grain size (r = −0.243), explained 25.2% of the variation; the second axis, which correlated most strongly with tree density (r = 0.716) and water content (r = −0.596), explained 8.3% of the total variation. Superimposing the clustering results of forest structure, sediment features, and macrobenthic communities showed that the relatively higher tree density and lower silt/clay content in the frequently flooded site (XF) were correlated with juvenile crab, shellfish larva, Dolichopodidae larva, Corophiidae, and Oligochaeta abundance; the higher tree density and silt/clay content in the high-elevation site (ZN) were correlated with *Phascolosoma arcuatum*, Assimineidae, and *Hyala bella* abundance; and the lower tree density in ZW, WZ, and FY was correlated most strongly with Capitellidae, Orbiniidae, Talitridae, and Grapsidae abundance. In summary, tree density and sediment silt/clay content were the main factors structuring the macrobenthic community in the *K. obovata* mangroves.

The results of the DistLM and dbRDA showed that the macrobenthic communities in the *A. marina* mangroves in BM and CK were very similar and mixed. The two axes explained 17.8% of the total variation in the macrobenthic communities in the *A. marina* mangroves (Figure 5b). The first axis, which correlated most strongly with organic content (r = 0.608) and the density of pneumatophores (r = −0.645), explained 11.6% of the variation; the second axis, which correlated most strongly with water content (r = 0.830) and grain size (r = 0.410), explained 6.2% of the total variation. Superimposing the clustering results of forest structure and sediment features with those for the macrobenthic communities showed that a lower pneumatophore density and higher organic and water content in the low-elevation site (BD) correlated most strongly with Stenothyridae, *Iravadia quadrasi*, and *Acteocina koyasensis* abundance; the higher pneumatophore density and greater grain size in FY correlated with Naticidae abundance; the higher pneumatophore density and lower water content in BM and CK correlated most strongly with *Phascolosoma arcuatum*, Nereididae, *Hyala bella*, Nemertea, Capitellidae, Amphipoda, and Ampharetidae abundance. In summary, pneumatophore density and sediment grain size were the main factors structuring the macrobenthic community in the *A. marina* mangroves.

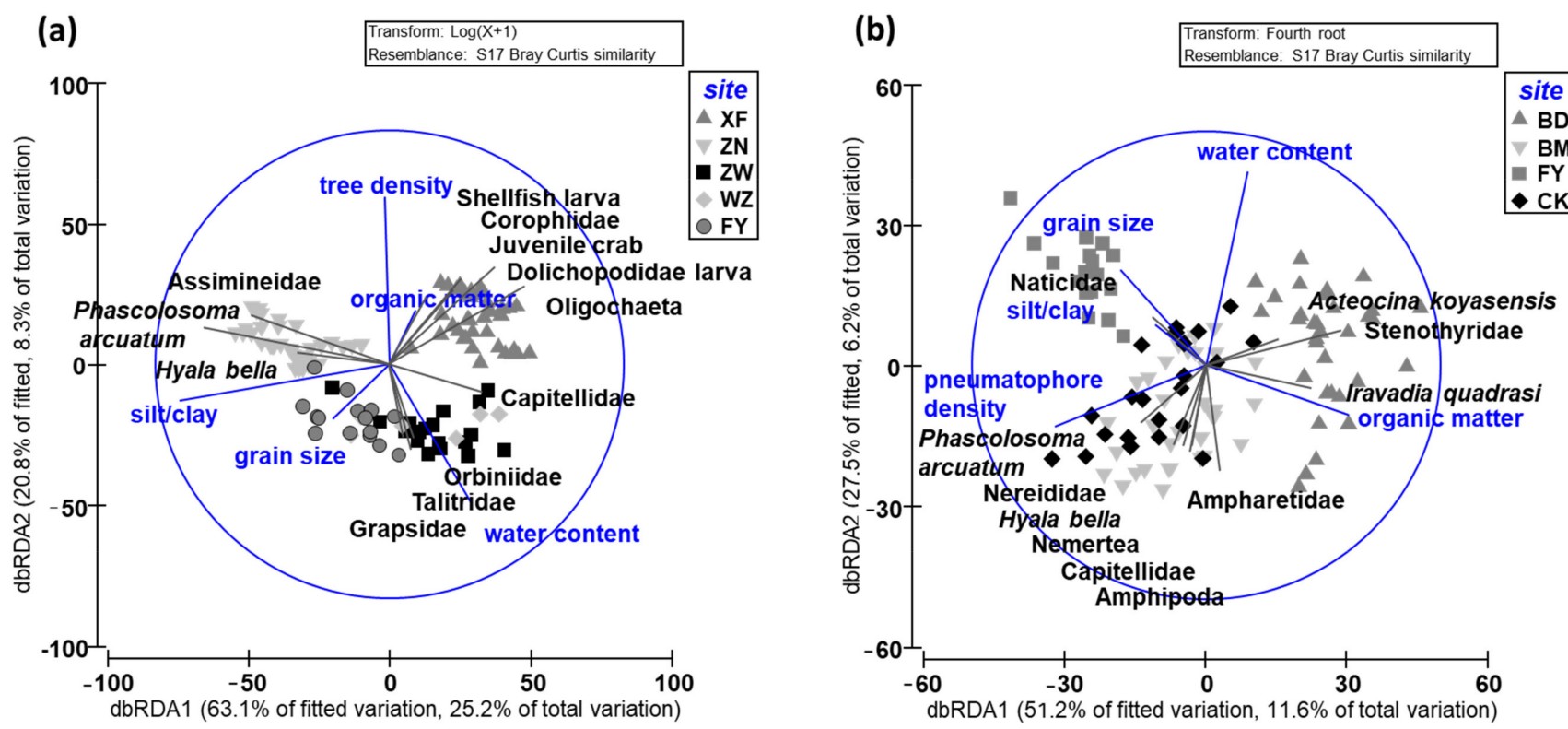

**Figure 5.** DistLM and dbRDA plots showing the relationships between stand structure, sediment features, and taxon density in the macrobenthic communities collected from (**a**) *Kandelia obovata* and (**b**) *Avicennia marina* mangroves of the four sites (XF, ZN, BD, and BM) sampled in this study. ZW, Huang (2017) [50]; WZ, Huang (2018) [51]; FY, Kuo (2016) [52]; CK, Yu and Lin (2015) [53]; FY, Kuo (2016) [52]. The locations of these study sites are shown in Supplementary Figure S1.

## 4. Discussion

### 4.1. Comparison between Mangrove Forests and Mudflats

In this study, crab density was significantly lower in all the mangrove forests than in the mudflats. A similar pattern was also observed by Chen et al. [54] in the Siangshan mangrove ecosystem in Taiwan. Sesarmidae crabs were the most and almost the only dominant crab species in all the mangrove forests. This crab species was also observed in most of the mudflats. In XF and ZN, the density of Sesarmidae crabs in the mangrove forests and mudflats was comparable. In the mudflats, however, the dominant crab species varied from site to site. The lower density of crab species in mangrove forests can be attributed to the preference of most crab species for mudflats over mangrove forests [55]. A greater density of Sesarmidae crabs was observed in *K. obovata* mangroves than in *A. marina* mangroves, as they generally feed on propagules of *K. obovata* [56].

Similar to the crab community composition, the community composition of macrobenthos other than crabs in the mudflats appears to be more dispersed than those in the adjacent mangrove forests. However, the macrobenthic communities in the mudflats were not richer in taxa or more diverse than those in the mangrove forests. Our results were contrary to some prior studies [57,58] but consistent with the observations of Checon et al. [59]. In this study, a variety of taxa, such as polychaetes, bivalves, gastropods, tanaids, and amphipods, existed only in mangrove forests. However, two species of gastropods were observed only in the mudflats. This suggests that the mangrove forests were a restricted habitat for some macrobenthic taxa, as indicated by Lin et al. [60], so that the macrobenthic community was more constrained to the mangrove forests than to the mudflats. The more constrained macrobenthic community composition in the mangrove forests than in the mudflats possibly resulted from the higher organic matter and lower sediment temperature and shelter provided by mangrove stand structure [23,61]. The epifauna living on the exposed mudflats are more vulnerable to the direct impacts of tidal waves or flooding streams, which might cause a higher risk of washout [62] and a rapid shift in the composition of the macrobenthic community. Nishijima et al. [63] indicated that flooding can erode the top 12 cm of sediments. As most infauna inhabit the top 5 cm of sediments [64], the infauna in the sediments of mudflats might be frequently affected by stronger hydrodynamic forces.

In contrast to the crab density, however, the density of other macrobenthos in the mangrove forests was not necessarily higher or lower than that in the adjacent mudflats. This lack of significance was unlikely to be caused by seasonal variation, as there was no seasonal variation in the macrobenthic density in the mangrove forests or mudflats. In only two of the four sites (ZN and BM) was the density of other macrobenthos in the mangrove forests significantly different from that in the mudflats. However, the results were inconsistent. In ZN, the density of other macrobenthos was significantly higher in the mudflats, whereas the density in BM was higher in the mangrove forests. Although there were distinct sediment features between the mangrove forests and the mudflats, our results suggest that not only sediment features [20,21] but also other factors discussed later might be involved in regulating the macrobenthic density in the forest mangroves.

The potential factors structuring macrobenthic communities in mangroves can be revealed by MDS grouping of all the communities in the adjacent mudflats and mangrove forests together (Figure 6). The MDS results showed that there were distinct macrobenthic communities at each site, although the macrobenthic communities could be further separated into the groups of *K. obovata* and *A. marina* mangroves and into mangrove and mudflat groups. This suggests that the local environment was the primary determinant in structuring the macrobenthic community at each site. For example, the density was lowest in the frequently flooded XF, where both communities in the mangrove forests and mudflats were dominated by Oligochaeta due to the very low water salinity (0.93) in this area [65]. Larvae of Dolichopodidae were also abundant in XF, as they prey upon oligochaetes [66]. The low water salinity may restrict the distribution of some macrobenthic taxa in XF. The dominance of Sipuncula *Phascolosoma arcuatum* in both mangrove forests and mudflats of high-elevation ZN suggests that *Phascolosoma arcuatum* prefers a high-elevation

habitat with relatively low water content and sediments with a small grain size. Snails (Stenothyridae and *Iravadia quadrasi*) were the dominant taxa in the mangrove forests of BD, which can reflect the high benthic Chl *a* concentration and longer submersion time of the sediments [67].

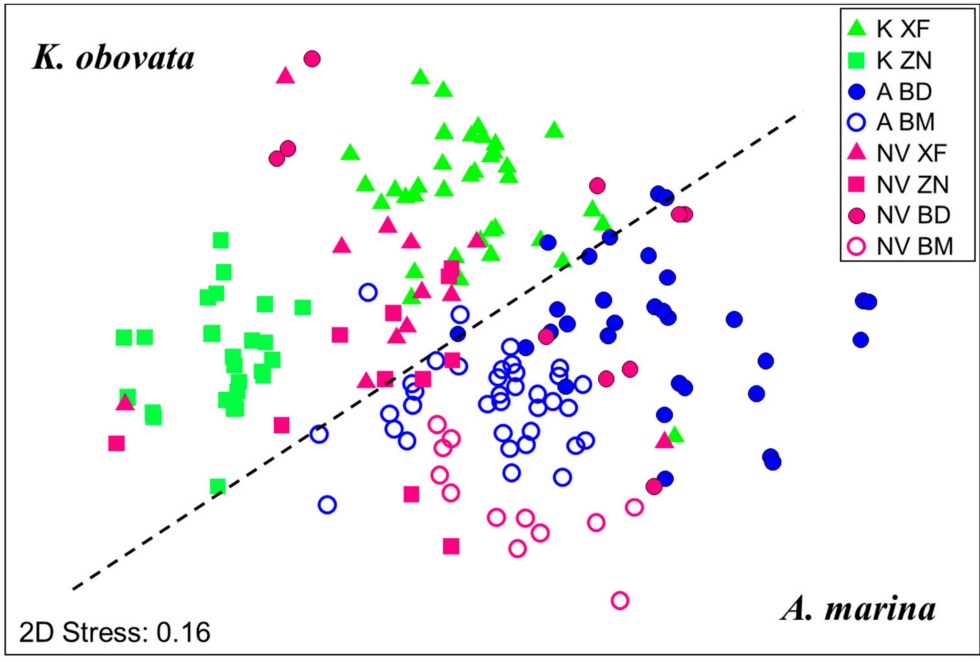

**Figure 6.** The MDS ordination of the macrobenthic communities collected from the four mangroves (K, *K. obovata*; A, *A. marina*) and nonvegetated mudflats (NV) of the four sites (XF, ZN, BD, and BM) on the western coast of Taiwan.

### 4.2. Comparison among Mangrove Forests

Thilagavathi et al. [68] suggested that the diversity of macrobenthic communities can be used to assess the health of mangrove ecosystems. In the *K. obovata* mangroves, the macrobenthos were more diverse in XF than in ZN, although the density was higher in ZN than in XF. In the *A. marina* mangroves, the taxon richness, diversity, and density of macrobenthos were higher in BM than in BD. Our results show that the diversity of macrobenthic communities primarily reflects the local conditions of sediment features or stand structure. The DistLM and dbRDA results show that more diverse macrobenthic communities in *K. obovata* mangroves (XF) occurred in sediments with smaller grain sizes and lower silt/clay contents, while the tree density was high. For *A. marina* mangroves, more diverse macrobenthic communities (BM) were observed in the mangroves with higher pneumatophore densities. Combined with the results of more constrained macrobenthic community composition in the mangrove forests than in the mudflats, it is clear that stand structure (i.e., tree or pneumatophore density) was a main factor structuring the diverse macrobenthic communities in the mangroves of both species.

Data on the macrobenthic density from other *K. obovata* (Supplementary Table S5) and *A. marina* (Supplementary Table S6) mangroves in Taiwan and the Indo-Pacific were combined to assess the relationships between forest stand density and macrobenthic density. For *K. obovata*, there were significantly negative relationships between tree density and total macrobenthic and infaunal densities (Figure 7a,b). There was no clear relationship between tree density and epifaunal density (Figure 7c). This suggests that the tree density of *K. obovata* had more negative impacts on infauna, particularly polychaetes (Figure 7d), than on epifauna. The macrobenthic density in the mangrove forests of XF and ZN was low compared to the density of other sites. One possible explanation for this phenomenon might be the high tree density at both sites. The greater the tree density is, the greater the

root biomass is, which may reduce the drilling behavior of infauna [19]. The decreased light intensity caused by the higher tree density and canopy shading might also result in a lower benthic Chl *a* concentration in the sediments, which may reduce the availability of food sources of herbivores. This may be the reason that the herbivore Assimineidae was more abundant in the mudflats than in the mangrove forests of XF and ZN.

In *A. marina* mangroves, there was no significant relationship between tree density and macrobenthic density. However, there was a significantly negative relationship between pneumatophore density and the density of epifauna, particularly gastropods (Figure 8). Penha-Lopes et al. [67] also observed a decreasing trend in the density of the gastropod *Terebralia palustris* with increasing mangrove tree density after the tree density reached a threshold. Prior studies have indicated the beneficial effects of pneumatophores on epifauna [23,24]. However, Bishop et al. [69] indicated that pneumatophore density did not have any impact on macrobenthos density, possibly because only samples of large epifauna >5 mm were collected in their study. In this study, no clear relationship between pneumatophore density and total macrobenthic density and infaunal density was detected (Figure 8a,b). However, there was a significantly negative relationship between pneumatophore density and epifaunal density (Figure 8c). It appears that a higher density of pneumatophores caused stress to epifauna, particularly gastropods (Figure 8d). While the pneumatophore density of *A. marina* mangroves was higher in BM than in BD, the sediment silt/clay content was higher, but the benthic Chl *a* concentration was lower in BM than in BD (Table 1). It is likely that a higher density of pneumatophores enhanced the deposition of suspended matter [70] and reduced the biomass of benthic microalgae [71], which are a food source for herbivores. Consequently, the density of *Acteocina koyasensis* and Stenothyridae feeding on benthic microalgae was higher in BD than in BM.

It has been shown that sea-level rise caused by climate change may affect the growth and distribution of mangroves [72–74]. Large-scale planting of mangroves has been applied as a mitigation strategy for mangrove loss [34]. The change in mangrove vegetation would directly affect the abundance and community composition of macrobenthos in the tidal flats and indirectly change the coastal biogeochemical cycles through the function of macrobenthos as ecological engineers [75,76]. However, the response of macrobenthic abundance and community composition to mangrove vegetation was inconsistent. In addition, local environment was the primary determinant in structuring the macrobenthic community. Therefore, the use of macrobenthic community as a health index for mangrove ecosystems or the prediction of the shift of macrobenthic community in response to mangrove loss or planting should take local conditions into consideration.

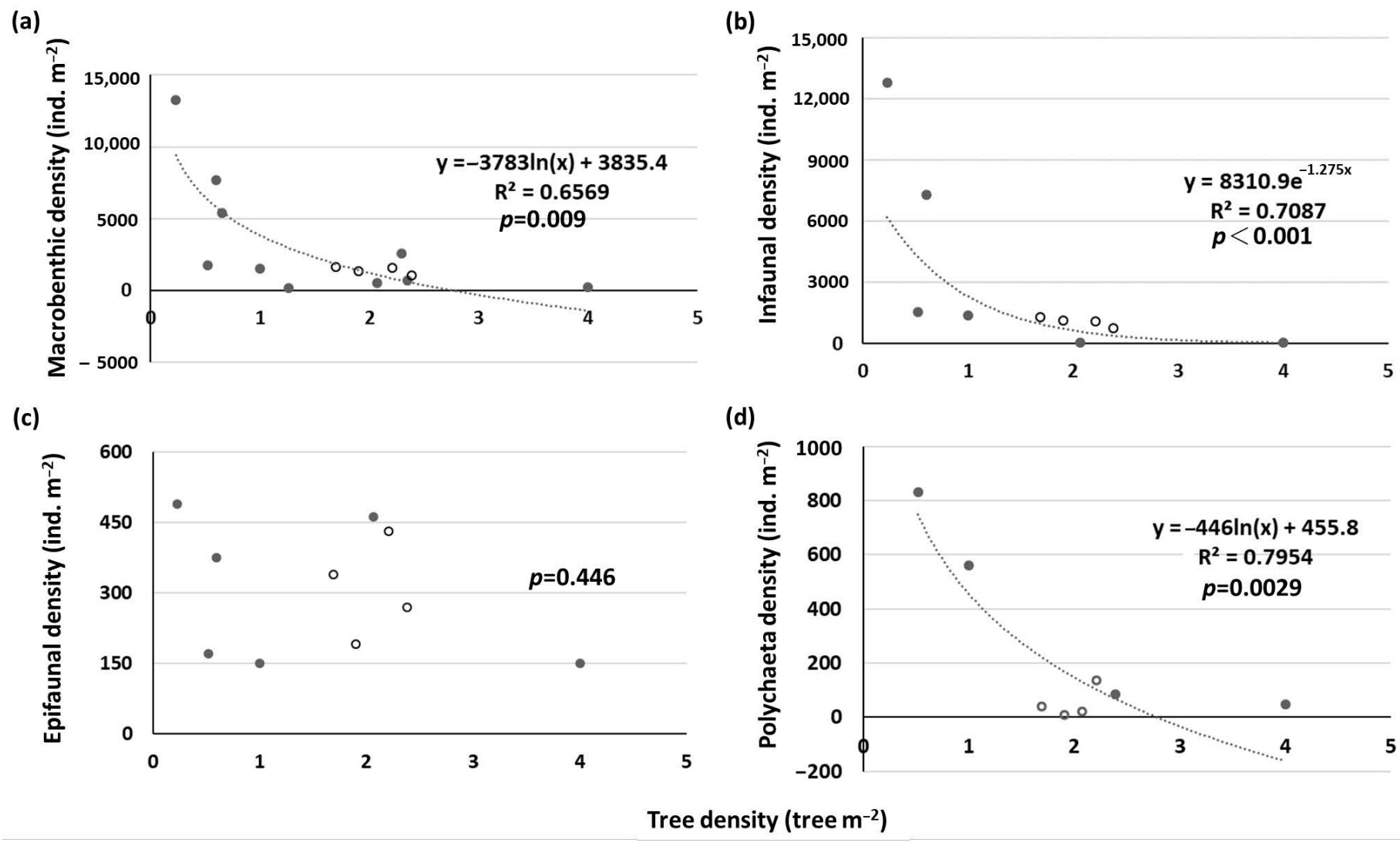

**Figure 7.** Relationships between the densities of (**a**) macrobenthos, (**b**) infauna, (**c**) epifauna, and (**d**) polychaetes and the tree density of *Kandelia obovata*. White dots: this study; black dots: prior studies.

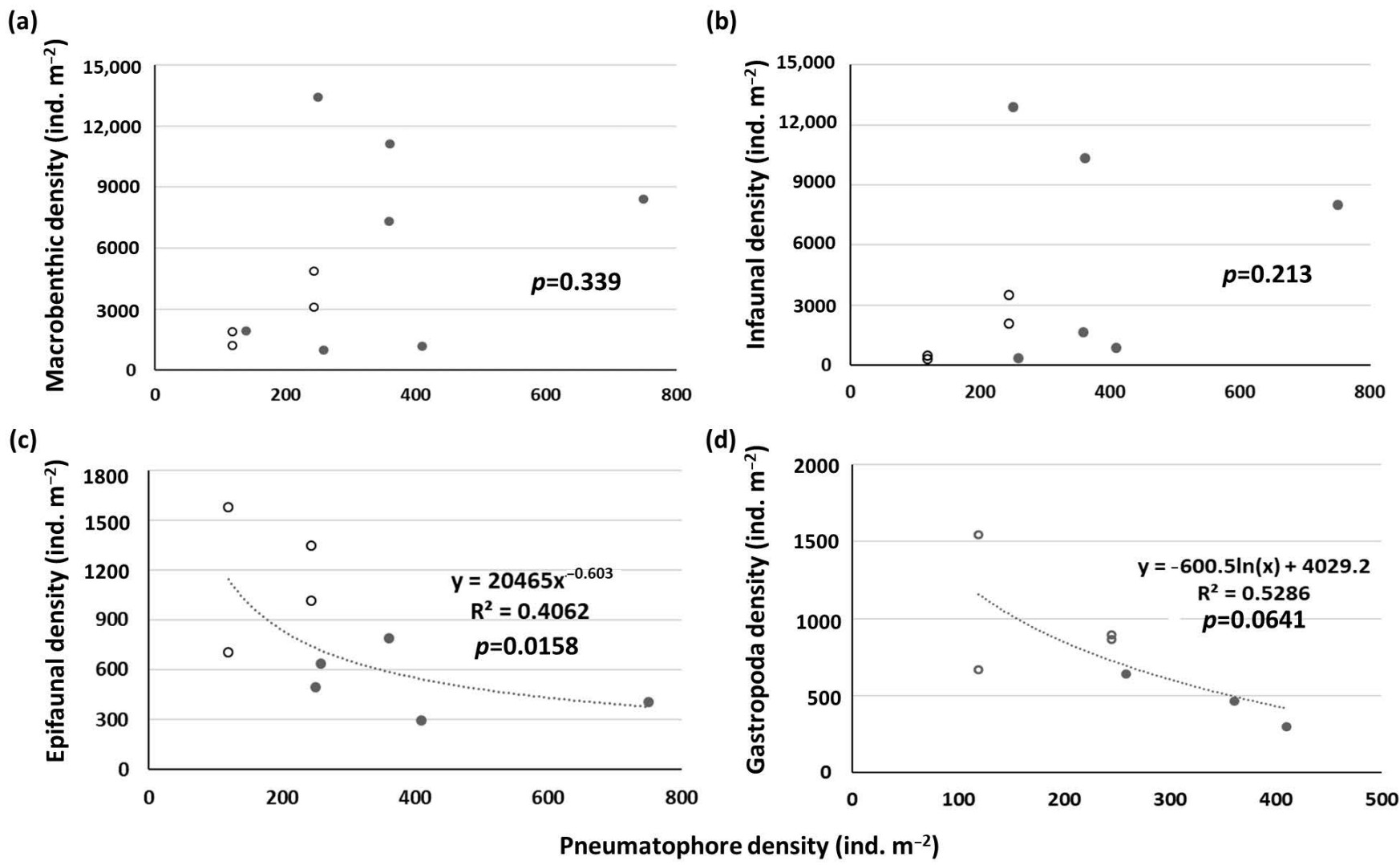

**Figure 8.** Relationships between the densities of (**a**) macrobenthos, (**b**) infauna, (**c**) epifauna, and (**d**) gastropods and *Avicennia marina* pneumatophore density. White dots: this study; black dots: prior studies.

## 5. Conclusions

Our results showed that mangroves are critical habitats for coastal macrobenthic communities. There were distinct macrobenthic communities in each sampled mangrove forest and adjacent nonvegetated mudflat. Although the crab density was always lower in the mangrove forests than in the mudflats, some polychaetes, bivalves, gastropods, tanaids, and amphipods existed only in the mangrove forests. In addition, the macrobenthic communities in the mudflats tended to fluctuate more than those in the mangrove forests, possibly due to the higher organic matter and lower sediment temperature and shelter provided by the stand structure. This suggests that stand structure was a main factor structuring the macrobenthic communities in the mangrove forests.

However, the response of macrobenthic density in the mangrove forest to stand structure differed from that of the macrobenthic community. High stand density may reduce the density of the macrobenthic community in mangrove forests. Together, the data from the present study and the relevant literature show that there were negative relationships between stand density and macrobenthic density. We further found that the effects of different stand structures of mangrove species on macrobenthic taxa differed. *K. obovata* possesses prop roots, so the tree density had more negative impacts on infauna, such as polychaetes. However, the density of pneumatophores of *A. marina* was found to be negatively correlated with the density of epifauna, such as gastropods.

In total, the response of macrobenthic abundance and community composition to mangrove vegetation was inconsistent. We reason that mangrove forests are critical habitats for the macrobenthos in the mudflats. However, if mangrove tree density is high, we predict that macrobenthic density will decrease. This suggests that at some intermediate level of mangrove tree density, where there are enough mangrove trees to harbor macrobenthic community but not enough trees to reduce density greatly, the management of mangroves can be achieved in an optimal way for macrobenthic community.

**Supplementary Materials:** The following are available online at https://www.mdpi.com/article/10.3390/f12101403/s1, Table S1: List of taxa in the macrobenthic community identified in the four sites on the western coast of Taiwan and their corresponding living and feeding habits based on Cai [1] and the World Register of Marine Species (WoRMS) [2]., Table S2: Analysis of the difference in macrobenthic variables between the mangrove forests and nonvegetated mudflats at the four sites on the western coast of Taiwan by Student's *t*-test (*t*-test) or the Wilcoxon rank-sum test (Wilcoxon). The bold values indicate significant differences ($p < 0.05$), Table S3: Analysis of the seasonal and site effects on the macrobenthic variables in the mangrove forests on the western coast of Taiwan by the Kruskal-Wallis test. Dunn's test was used to determine which season or site showed a significant difference. The bold values indicate significant differences ($p < 0.05$), Table S4: Analysis of the seasonal and site effects on the macrobenthic variables in the nonvegetated mudflats on the western coast of Taiwan by two-way ANOVA (F value) or the Kruskal-Wallis test (H value). Tukey's or Dunn's test was used to determine which season or site showed a significant difference. The bold values indicate significant differences ($p < 0.05$), Table S5: Comparisons of the densities of Kandelia obovata and macrobenthos observed in this study and other studies. FY: Fangyuan, ZN: Zhunan, XF: Xinfeng, GD: Guandu, ZW: Zhuwei, and WZ: Wazihwei. The locations of the study sites in Taiwan are shown in Fig. S1, Table S6: Comparisons of Avicennia marina tree and pneumatophore density and the density of macrobenthos in this study and other studies. CK: Chiku, BM: Beimen, BD: Budai, and FY: Fangyuan. The locations of the study sites in Taiwan are shown in Figure S1, Figure S1: Locations of the study sites on the western coast of Taiwan. From north to south: WZ, Wazihwei; ZW, Zhuwei; XF, Xinfeng; ZN, Zhunan; FY, Fangyuan; BD, Budai; BM, Beimen; CK, Chiku. GD (Guandu) overlaps with ZW. Among these sites, XF, ZN, BD and BM were sampled in this study.

**Author Contributions:** Conceptualization, H.-J.L.; data curation, S.-H.P., C.-W.H., C.-W.L. and S.-C.H.; formal analysis, C.-W.L. and C.-W.H.; funding acquisition, H.-J.L.; investigation, S.-H.P., C.-W.H., C.-W.L. and S.-C.H.; methodology, H.-J.L.; project administration, H.-J.L.; supervision, H.-J.L.; validation, C.-W.H. and H.-J.L.; writing, S.-H.P., C.-W.H., C.-W.L. and H.-J.L. All authors have read and agreed to the published version of the manuscript.

**Funding:** The study was granted by the Ministry of Science and Technology (MOST) of Taiwan (106-2621-M-005-005-MY3) and the "Innovation and Development Center of Sustainable Agriculture" from The Featured Areas Research Center Program within the Higher Education Sprout Project by the Ministry of Education (MOE) of Taiwan.

**Institutional Review Board Statement:** Not applicable.

**Informed Consent Statement:** Not applicable.

**Data Availability Statement:** Not applicable.

**Acknowledgments:** We are grateful for the support of the Ministry of Science and Technology (MOST) of Taiwan under grant no. 106-2621-M-005-005-MY3. This work was also financially supported in part by the "Innovation and Development Center of Sustainable Agriculture" from The Featured Areas Research Center Program within the Higher Education Sprout Project by the Ministry of Education (MOE) of Taiwan. The authors express their special thanks to all team members from Aquatic Ecosystem Lab of National Chung Hsing University, who helped during the field work.

**Conflicts of Interest:** The authors declare no conflict of interest.

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
