# Peer review of "Differential Response of Macrobenthic Abundance and Community Composition to Mangrove Vegetation"

_forests, doi:10.3390/f12101403_

Round 1

Reviewer 1 Report

Authors studies differential response of microbenthic abundance and community composition to mangrove vegetation. Authors have collected several data sets from different sites, seasonal data and related abiotic and biotic factors to explain infauna and epifaunal community composition. Overall study is quite impressive however overall manuscript is quite confusing and doesn’t reflect what is the main objectives of the study.  

A better story is needed to understand the whole research. I would suggest to see (1) seasonal response (2) site response and (3) environmental variables response to understand outcome of this study. Currently, everything is not clearly written. Based on above authors need to formulate hypothesis.

Line 12-17= Abstract started with mass plantation. It is not clear from the successive lines, where exactly this study was performed (Restored, intact or degraded mangroves). Study was done in four forests (what kind of forests are they, need to clarify).

Line 18-24= Community dispersion was higher in the mudflat but reason suggests that rich food supply, low temperature and shelter function are better in the mangrove areas. Also other relationships are negative. Need better explanation because statements are contradictory.

What is the implication of this study. I do not see a very solid conclusion. According to study conclusion, people should not plant restore mangroves or if natural regeneration is occurring what do you suggest. Overall abstract need to rewrite it doesn’t reflect the whole study such as justification, main findings and a solid conclusion.

Line 98-101 = Add references and year when and how this data was obtained.

Figure 1 is quite not clears and difficult to understand. Also in the PCA analyses seasons were also used. However, I do not see any results explaining seasons in the text from figure 1. (Why seasonal data were used?)

Figure 2 and 3 can be combined to understand results better. Also add significant ANOVA results in the figure 2a, 2b, 3a and 3b. same letters are not significantly difference.

Conclusion basically only contain results but I do not see any implication of this study. Some results are good but how can we use new information in terms of mangrove restoration.

Below are some recently published papers which is quite related to this study.

Cannicci, S., Lee, S. Y., Bravo, H., Cantera-Kintz, J. R., Dahdouh-Guebas, F., Fratini, S., ... & Diele, K. (2021). A functional analysis reveals extremely low redundancy in global mangrove invertebrate fauna. Proceedings of the National Academy of Sciences, 118(32).

Egawa, R., Sharma, S., Nadaoka, K., & MacKenzie, R. A. (2021). Burrow dynamics of crabs in subtropical estuarine mangrove forest. Estuarine, Coastal and Shelf Science, 252, 107244.

Reviewer 2 Report

Congratulations to the authors for their contribution. The manuscript is clear in all aspects and includes relevant information to readers interested on mangroves and associated macrobenthic communities. I just suggest very minor changes, e.g. Figure 1 should be somehow improved in order to make clear the text related to seasons. Figure 2,3: I suggest to use an uppercase for density (i.e. Density). Same for abbreviations (Win, Spr, Sum, Aut) (Figs 2,3).

Figure 4: Same comment, please use an uppercase for the first letter in Y-axis and abbreviations for seasons.

Round 2

Reviewer 1 Report

Authors have revised manuscript as per reviwers comments and happy to accepted.